# Single-Cell Analysis Dissects the Effects of Vitamin D on Genetic Senescence Signatures Across Murine Tissues

**DOI:** 10.3390/nu17030429

**Published:** 2025-01-24

**Authors:** Emilio Sosa-Díaz, Helena Reyes-Gopar, Guillermo de Anda-Jáuregui, Enrique Hernández-Lemus

**Affiliations:** 1Computational Genomics Division, National Institute of Genomic Medicine, Mexico City 14610, Mexico; sosadiazemilio12@gmail.com (E.S.-D.); hreyesgopar@northwell.edu (H.R.-G.); 2Center for Complexity Sciences, Universidad Nacional Autónoma de México, Mexico City 04510, Mexico; 3Investigadores por Mexico, Conahcyt, Mexico City 03940, Mexico

**Keywords:** vitamin D, cellular senescence, single-cell genomics, aging, murine model

## Abstract

**Background/Objectives:** Vitamin D (VD) plays a crucial role in age-related diseases, and its influence on cellular senescence (CS) could help clarify its function in aging. Considering VD’s pleiotropic effects and the heterogeneity of CS. **Methods:** we utilized single-cell RNA sequencing (scRNA-seq) to explore these dynamics across multiple tissues. We analyzed three murine tissue datasets (bone, prostate, and skin) obtained from public repositories, enriching for senescence gene signatures. We then inferred gene regulatory networks (GRNs) at the tissue and cell-type levels and performed two cell communication analyses: one for senescent cells and another for interactions between senescent and non-senescent cells. **Results:** VD supplementation significantly decreased senescence scores in the skin (*p* = $3.96\times 10^{-134}$) and prostate ($p=1.56\times 10^{-34}$). GRN analysis of the prostate revealed an altered macrophage–fibroblast regulatory relationship. In bone, distinct aging-related modules emerged for different bone lineages. In skin, contrary differentiation patterns between suprabasal and basal cells were observed. The main VD-modulated pathways were involved in inflammation, extracellular matrix remodeling, protein metabolism, and translation. VD reduced fibroblast–macrophage interactions in the prostate and skin but increased overall cellular crosstalk in bone. **Conclusions:** Our findings demonstrate that VD alleviates CS burden across tissues by modulating inflammation and metabolic processes and promoting differentiation. Key aging-related genes modulated by VD were linked to anabolism and cellular differentiation, suggesting VD’s potential for therapeutic interventions targeting age-related diseases.

## 1. Introduction

Vitamin D is a steroid hormone obtained by ingestion or photo-reaction in the skin cells [1]. This hormone is necessary in mineral and bone homeostasis, although new insights are proving vital in our understanding of the non-canonical effects of the vitamin, because it seems to exert action in maladies like cancer and age-related diseases [2].

Remarkably, vitamin D deficiency is very prevalent worldwide, and approximately a billion of the world’s population is at risk of presenting inadequate serum vitamin D concentrations [3]. This has caused massive public health efforts to establish interventions, such as enhancing the use of fortified food with vitamin D, although VD deficiency is still present in countries that have taken the said measures [4]. Consequently, physicians and researchers now have associated many diseases with these VD deficits depending on the magnitude of patient deficiency.

An important functional axis in aging alterations is cellular senescence. This phenomenon refers to replicative cell halt and continuous growth under determined stress or physiological signals like oncogenic stress, deoxyribonucleic acid (DNA) damage, or early embryonic development [5]. This process represents an alternative cell fate and has been viewed as a culprit of aging, mainly in chronic degenerative diseases such as diabetes, hypertension, and cancer [6]. Current research on this issue focuses on removing or harnessing the characteristics of these cells. As such, vitamin D points to antagonizing senescent cells with its anti-apoptotic, energy-demanding, and pro-inflammatory profile, while it promotes anti-proliferative and differentiation characteristics. Thus, vitamin D may be a candidate fit for cellular senescence-focused therapeutics, also known as senotherapeutics [7].

Cellular senescence has proven to be a challenging cellular state given its heterogeneity across cell types and trigger mechanisms, so there is no definite biomarker nor genetic signature that certainly predicts whether a cell is properly senescent [8]. Furthermore, vitamin D shows pleiotropism between tissues and even cell types, making it difficult to target its non-canonical effects [9]. Importantly, key tissues such as bone, prostate, and skin highlight the value of studying this relationship due to their distinct interactions with vitamin D and senescence. Low levels of its active form, 1,25-dihydroxyvitamin D, accelerate bone loss through oxidative stress and the accumulation of senescent osteocytes, leading to impaired remodeling and osteoporosis [10]. In the prostate, vitamin D has anticancer properties, reducing the risk of aggressive prostate cancer by influencing cell differentiation and senescence. Supplementation has shown promise in slowing tumor growth [11]. In the skin, vitamin D supports healthy aging by reducing inflammation and mitigating the effects of cellular senescence, promoting overall skin health and resilience [12]. To tackle these complex interactions, a more comprehensive approach is needed to fully elucidate vitamin D’s role within senescence.

Along these lines, single-cell approaches may be an exhaustive tool to thoroughly integrate the big picture of biological phenomena. In this particular case, single-cell RNA sequencing can be a valuable experimental tool to suggest cellular function [13]. This is to characterize cell populations, cell states in interventions, key biological pathways, and even differentiation pathways and cellular communication [14]. However, the downside of this method is that there are few experiments available given its high cost, technical challenges (like sample preparation), data complexity, and recent implementation in the market [15].

In this study, we investigate the relationship between vitamin D and cellular senescence using single-cell RNA sequencing, aiming to understand how vitamin D influences senescence across various tissues and cell types. By taking a non-canonical perspective on the aging process, we seek to reveal novel insights into how vitamin D might delay or modulate age-related changes. Ultimately, this work may inspire new research and potential interventions against aging-related diseases by harnessing vitamin D’s beneficial effects.

## 2. Materials and Methods

### 2.1. Data Acquisition

Research in the literature was conducted with the keywords *vitamin D* and single cell RNA seq in PUBMED and GoogleScholar with the aim to recruit the publicly available, single-cell RNA seq data of experiments using vitamin D. Six datasets were originally recovered. However, under further inspection, two used bulk RNA sequencing techniques, so they were excluded. Four datasets were selected for further data exploration [16,17,18,19]. Nevertheless, further analysis was performed only with the databases of mouse bone, skin, and prostate [16,17,18] given quality control results (see Section 2.4). It is worth noting that the bone tissue datasets are from eight healthy mice, the prostate dataset is from a PTEN-mouse, and the skin tissue datasets were collected from UV-irradiated mice.

### 2.2. Data Structure

The raw prostate dataset included a vehicle count matrix with features and barcodes with 5163 cells and 17,744 genes and a VD count matrix with features and barcodes with 7912 cells and 17,934 genes (GSE164858). The raw skin dataset included an annotated vehicle count matrix with 17,666 genes across 9081 cells and an annotated VD count matrix with 17,257 genes across 6200 cells (GSE173385). The raw bone dataset included a vehicle count matrix with its features and barcodes with 18,208 genes in 4629 cells and a VD count matrix with features and barcodes with 19,008 genes across 3914 cells (GSM6820990). The integrated prostate dataset included 13,075 cells and 18,535 genes in the expression matrix; the bone dataset included 8543 cells and 19,516 genes; and the skin dataset included 15,281 cells and 18,545 genes.

Each dataset’s experimental structure is thoroughly detailed in its respective source article [16,17,18], and we encourage readers to consult these references for comprehensive descriptions. Briefly, the prostate dataset experiment involved 30 Pten(i)pe-/- mice, 9 months post-Pten inactivation, which were randomly assigned to receive either 100 μL of Gemini-72 (a potent calcitriol analog) or a vehicle daily for 1, 2, or 3 weeks. Ten mice were sacrificed each week to evaluate prostate outcomes [16]. The bone dataset experiment used 12 seven-week-old male C57BL/6J mice divided into untreated and treated groups, with the treated group receiving 2.5 μg/kg calcitriol daily for three days. Femurs were collected one day after the final calcitriol dose, following euthanasia [17]. The skin dataset experiment included 9 five- to six-week-old male C57BL/6 mice, divided into three groups: one exposed to normal light and two to UVB (313 nm, 300 mJ/cm^2^). Of the UVB-exposed groups, one received vitamin D injections (50 ng). Mice were treated three times weekly for 50 min over three weeks [18].

### 2.3. Senescence-Associated Gene Sets

Three senescence-associated gene sets were retrieved for enrichment: SENmayo, with 119 genes mainly associated with the senescence transcriptome [20]; Geneage, with 177 genes associated with overall aging [21]; and Cellage, with 949 genes associated with cellular senescence specifically [22].

### 2.4. Preprocessing

Statistical analyses were performed using R Statistical Software (v4.3.2; R Core Team 2023) and all plots were made with the ggplot2 package v3.5.1 [23]. The Seurat package v5.1.0 was used for the management of the scRNAseq data [24]. First, all datasets were subjected to quality control evaluating the number, count, number of genes, and the percentage of mitochondrial genes. Threshold values were set as the number of genes to be greater than 200 and mitochondrial RNA transcripts of less than 7.5%. Subsequently, data were normalized with LogNormalize, scaled by a factor of 10,000, and further scaled for identifying relevant variable genes. Then, dimensionality reduction using principal component analysis (PCA) and Uniform Manifold Approximation and Projection (UMAP) using the the first 10 first dimensions was performed. Finally, data were clustered using the KNN algorithm with PCA Euclidean distances.

#### 2.4.1. Annotation

Cellular-type annotation was carried out with the SingleR package v2.8 [25] using mouse prostate [26], skin [27], and bone [28] atlases as reference.

#### 2.4.2. Integration

For batch effect and condition vs. control (VD vs. vehicle) integration, the Harmony algorithm included in the Seurat package [24] was applied. Finally, Uniform Mannifold Approximation and Projection (UMAP) plots were obtained for cluster and cell-type visualization. Annotation and integration raised questions regarding a cell type identified as immune in the hanai (bone) dataset. Under further investigation, they were identified as myeloid precursors, consistent with what was reported in the original article of the dataset. Given the interest in studying cortical bone only and not bone marrow cells, they were removed from the dataset.

### 2.5. Functional Enrichment

Biological function enrichment of the processed datasets was performed using the AUCell algorithm [29] via the Escape package v2.2.2 [30] with the three senescence-associated gene sets. Here, for gene conversion from human to mouse, the biomart package v2.62.0 was used [31]. To see if there was a difference between enrichment scores for vitamin D and vehicle groups, a pairwise Wilcoxon test for each cell type and for overall tissue was performed [32] with a significance of *p* < 0.05.

### 2.6. Gene Regulatory Network Analysis

#### 2.6.1. Network Processing

Gene regulatory networks (GRNs) were made using the hdWGCNA package [33]. GRNs were acquired for the 3 overall tissues using a pseudobulk approach. Briefly, all the UMIs were added according to the cell type and the intervention (vitamin D vs. control) in a gene expression matrix. The matrix was normalized and subsequently the soft powers and the co-expression matrix were obtained. Finally, network topological variables like module eigengene connectivity were calculated for analysis. It is worth noting that, given the huge dropout in sc RNA seq data, the fraction of cells that a gene needs to be expressed in order to be included is set to 5%.

#### 2.6.2. Network Analysis

Upfront analysis involved identifying hub genes and module eigengenes via topological analysis of the networks. Afterwards, differential module eigengene (DME) expression between vitamin D and vehicle was assessed in every tissue in every cell type with more than 70 cells for significant statistical testing. Subsequently, to evaluate which genes were involved in the senescence phenomenon, an overlap analysis was performed between module genes and genetic senescence signatures via the ComplexUpset package v1.3.3. Then, to assess the relevance of the overlapped genes in the modules, module eigen-centrality was evaluated for the overlapped genes. Later, GO enrichment of the modules and an intersection of the overlapped genes with said enrichment was carried out to evaluate in which biological processes of the module the senescence genes participated. This was performed using the Enrichr package v3.2 [34]. Finally, the DME of the overall tissue was obtained for each module. Moreover, to properly address the cellular response to VD in each module, the cell types with the minimal and maximal log fold change for each module were assessed.

### 2.7. Cellular Communication

Data analysis was performed using two complementary approaches, by looking at the interactions between senescent cells exclusively and the interactions between senescent cells and other cell types in the tissue. For this reason, cells of each tissue and intervention (VD vs. vehicle) with a SENmayo enrichment score in the 90th quantile were clustered in a cell group named senescent_cells.

First, for each analysis, differential cellular interaction number and strength were investigated. Then, differential pathway analysis was performed. This was accomplished in three ways: by manifold learning with functional analysis, by pattern recognition of dominant communication, and via information flow differences. All these analyses were performed with the cellchat package v2.1.0 [35].

## 3. Results

### 3.1. Cell Annotation UMAP Between VD and Vehicle

First, we used Uniform Manifold Approximation and Projection (UMAP) to project high-dimensional gene expression data into two dimensions (shown on the x- and y-axes), making it easier to visualize how cells group based on similarity. We included 30 principal components to capture the major sources of variation and set the clustering resolution to 1 (which controls how finely we distinguish clusters). In each subpanel, the left plot shows vitamin D-treated samples, and the right plot shows vehicle-treated controls. We used different colors to indicate distinct cell clusters (e.g., epithelial, immune, and connective tissue cells). Cells that appear close together share more similar gene expression patterns. We observed shifts in cluster distribution and density between treatments. A common trend in all tissues was the greater quantity of cells in VD than in control (Figure 1). In the prostate (Figure 1A), the clusters in the VD sample appeared more interconnected than those in the vehicle sample. In the skin (Figure 1B), the epidermal (green) clusters appeared more separated from the hair follicle (pink) clusters in the vitamin D-treated sample, and each cluster was more interconnected than in the control. Furthermore, the macrophage cluster occupied a smaller area in the VD samples than in the control. In the bone (Figure 1C), chondrocyte, osteocyte and vascular smooth muscle clusters seem more abundant in the VD group.

### 3.2. Functional Enrichment

Then, we conducted gene set enrichment to see whether senescence-associated genes were more expressed in the control or in the vitamin D group. Figure 2 shows violin/box plots of SENmayo enrichment scores for vitamin D (VD) versus vehicles in different tissues. In the prostate (Figure 2A), SENmayo scores were significantly reduced in the VD group, with a median of 0.16 compared to 0.17 in the vehicle group. The *p*-value was less than $1.56\times 10^{-134}$, and the rank-biserial correlation was negative 0.12. In the bone (Figure 2B), SENmayo scores showed a slight increase in the VD group, with a median of 0.17 compared to 0.16 in the vehicle group. The *p*-value was 0.02, and the rank-biserial correlation was 0.04. In the skin (Figure 2C), SENmayo scores were significantly reduced in the VD group, with a median of 0.11 compared to 0.12 in the vehicle group. The *p*-value was less than $3.96\times 10^{-134}$, and the rank-biserial correlation was negative 0.26. Additionally, cellage scores decreased in the prostate but increased in the skin and bone for the VD group. Geneage scores decreased in the prostate and skin but showed no significant changes in the bone (Appendix A).

Cell-type-specific analysis revealed distinct changes across tissues. In the skin, SENmayo scores were significantly reduced in suprabasal cells one and two, as well as in hair follicle cells, while macrophages showed increased scores. In the prostate, T cells, B cells, macrophages, and fibroblasts exhibited significantly reduced SENmayo scores in the VD group compared to the vehicle. In the bone, SENmayo scores increased in chondrocytes, osteocytes, and osteoblasts, while scores decreased in adipocytes (Appendix A).

### 3.3. Gene Regulatory Network Analysis

Afterwards, we developed gene regulatory networks to study how vitamin D affects the expression of senescence-associated genes. Gene regulatory networks group genes that work together into “modules”, which are sets of genes that share similar patterns of activity. This allows us to study changes in expression at a broader level, rather than focusing on individual genes. To analyze these changes, we used a method called pseudobulk analysis, where data from individual cells of the same type are combined to create an “average” profile for each cell type. Using this approach, we identified between 32 and 42 modules for each tissue. In Figure 3, we explored an integrative analysis of gene modules under vitamin D treatment compared to the vehicle control. We examined their differential expression, their association with senescence-related genes, and their connectivity. We also analyzed their involvement in biological processes. This analysis was performed at two levels: the cellular level and the overall tissue level. Remarkably, in prostate tissue, fibroblasts tend to have downregulated modules, while immune cells, particularly macrophages, tend to have upregulated modules (Figure 3A). In the skin, we can see a clear pattern of upregulation across all cell types except in basal and cycling keratinocytes, where modules are heavily downregulated. (Figure 3C). Finally, non-parenchymal cells of the cortical bone (like tenocytes, myocytes, and chondrocytes) had their modules upregulated, while parenchymal cells (like osteoblasts, osteocytes, and osteoclasts) were downregulated. Notably, only green and pink modules are statistically significant among parenchymal cells, and they are upregulated in almost every other cell type (Figure 3B).

Regarding overlap analysis in the prostate (Figure 3D), the median connectivity of senescence-associated genes was 0.40 (range: 0.24–0.55). The differential module eigengene (DME) analysis revealed two trends. First, there is fibroblast downregulation and macrophage upregulation. Second, there is an overall downregulation of the modules, with dark red and blue modules showing no significant changes in the overall DME. The blue module was downregulated in myeloid leukocytes but upregulated in lymphoid leukocytes as seen in the left barplot. Within the senescence genes of the blue module, myeloid immunity, degranulation, and dendritic cell chemotaxis were downregulated, while B-cell differentiation, leukocyte aggregation, natural killer T-cell chemotaxis, and NF-kB activity were upregulated. This suggests that vitamin D increases macrophage activity while reducing their aggregation. It also reduces lymphoid degranulation while enhancing their chemotaxis and differentiation. Vitamin D also appears to influence other biological processes in the prostate. It upregulated DNA repair, replication, transcriptional regulation, extracellular matrix organization and assembly, protein ubiquitination and neddylation, the insulin growth factor pathway, mitophagy, apoptosis, and the estrogen response. These effects were primarily observed in macrophages, while the same processes were reduced in fibroblasts. In bone, the median connectivity of senescence-associated genes was 0.36 (range: 0.27–0.47), which is lower than in the prostate. The barplot shows that three modules are upregulated—magenta, salmon, and blue—while the rest are downregulated (Figure 3E). Regarding biological processes in the upregulated modules, the magenta module is associated with the unfolded protein response (UPR) and gene expression pathways, particularly in chondrocytes. The salmon module is involved in the interferon pathway, though no specific cell type shows significant changes in this module. The blue module is linked to rRNA regulation and macroautophagy, which are upregulated in chondrocytes but downregulated in tenocytes. For the downregulated modules, an opposing trend is observed between tenocytes and myocytes. In tenocytes, transcription inhibition in response to stress and protein degradation are reduced, while these processes are upregulated in myocytes. Additionally, vitamin D appears to inhibit vascular reorganization in myocytes. In skin, the median connectivity of senescence-associated genes was 0.29 (range: 0.24–0.42). The DME barplot reveals an opposing trend between suprabasal and basal cells, with modules in suprabasal cells being upregulated and those in basal cells downregulated, except for the green module, which is downregulated overall (Figure 3F). For biological processes, the only upregulated module (blue) is associated with angiotensin signaling activation, skin barrier establishment, and epithelium development. The remaining modules show overall downregulation in cytoplasmic translation, extracellular matrix (ECM) reorganization, oxidative phosphorylation, and autophagy. However, these processes are cell type-dependent. For example, the tan, dark grey, and brown modules are downregulated across all significant cell types and are linked to translation, stress-mediated response, and peptidyl-serine auto-phosphorylation. In contrast, the yellow, light cyan, blue, and cyan modules are upregulated in all significant cell types and are involved in autophagy, oxidative phosphorylation, and skin barrier function. The remaining modules follow the opposing basal–suprabasal trend observed in the DME barplot.

### 3.4. Cell Communication Analysis

Information flow between cells is essential to coordinate and maintain biological processes required for organ homeostasis and, ultimately, the survival of the individual. Given the distinct secretory profile of senescent cells, it is important to examine how they influence each other and other cell types. To address this, in Figure 4 and Figure 5, we evaluated the primary interactions, pathways, and differential relationships between VD and control tissues.

In the prostate, we found that senescent cell communication increased for interactions where fibroblast Fbn1 acted as the source and luminal Clu and B cells as receptors. In contrast, interactions from luminal cells and fibroblasts to macrophages decreased (Figure 4A). For the functional analysis, we identified four clusters of pathways based on similarity. JAM and BMP, the pathways with the greatest divergence, grouped into entirely distinct clusters, with vitamin D pathways in violet and vehicle pathways in blue (Figure 4D). In Figure 4G, we observed that vehicle-enriched pathways emphasized inflammation (CD34, IL4, TNF, COMPLEMENT) and migration (SELPLG, ICAM, FN1). In comparison, vitamin D-enriched pathways focused on more complex immune modulation (IL6, IL10, CD200), enhanced adhesion (CLDN, OCLN, NECTIN), and growth (PDGF, VEGF, IGF).

In bone, we found that senescent cell communication decreased only in interactions involving pericytes as receptors, while interactions for the rest of the senescent cell types increased. Senescent chondrocytes showed the highest upregulation (Figure 4B). In the functional analysis, we identified SEMA6, JAM, and VEGF as the pathways with the greatest divergence (Figure 4E). Vitamin D upregulated pathways associated with vascularization (EGF), structural integrity (collagen, fibronectin, tenascin), inflammatory response (TNF, CD80, SLURP), neural regulation (UNC5, SEMA6, SEMA7), and metabolic regulation (ENHO, apelin, PTH, and melanocortin). Conversely, vitamin D downregulated pathways involved in bone formation and maintenance (IGF, SPP1, periostin, BSP, and RANKL), immune chemotaxis (CCL, CXCL, MIF), neural development (NGF, NTS, NRXN, and SEMA4), cell adhesion and integrity (NCAM, CEACAM, CDH, NECTIN, and CLDN), and vascular regulation (ANGPTL and EDN) (Figure 4H). In skin, we found that fibroblasts, particularly dermal papilla fibroblasts, were the main upregulated senders, while the germinative layers of the hair follicle showed the most downregulation. For the receivers, myocytes and macrophages were notably downregulated, whereas basal cells were primarily upregulated (Figure 4C). Functional analysis identified ADGRG and NOTCH as the most altered pathways (Figure 4F). In the signaling flow analysis, we observed that vitamin D-enriched pathways promoted cell adhesion (DESMOSOME, LAMININ, CDH1, CLDN) and differentiation and development (EPGN, WNT, ADGRA). In contrast, vehicle-enriched pathways included those related to inflammation (COMPLEMENT, GAS, IL1), adhesion, and angiogenesis (PECAM, VCAM) (Figure 4I).

In Figure 5, we explore how vitamin D treatment influences cellular communication involving senescent cells, defined as those with the top 10 percent SENmayo enrichment scores, with other cell types across the prostate, bone, and skin tissues.

Vitamin D treatment in the prostate increases the activity of specific sender cell types, such as fibroblasts (cxcl9 and fbn1) and mesothelial cells, while it reduces activity in other fibroblast populations and senescent cells (Figure 5A). The functional analysis identifies SPP1 and HSPG1 as the pathways most affected by the treatment (Figure 5D). Under vitamin D, pathways associated with extracellular matrix remodeling (NECTIN, ANGPTL, GAP, TENASCIN, FN1), differentiation, chemotaxis (VCAM, ncWNT, CX3C), and inflammation (CD40, TNF, PTN) are enriched, while the vehicle condition is predominantly linked to antigen presentation (MHCI) (Figure 5G). For sender information, bone evaluation shows that cortical parenchymal bone cell (periosteum, osteocytes, osteoblasts) interactions were upregulated by VD, with fibroblast, senescent cell, and chondrocyte interactions being the most downregulated. Receiver information was markedly upregulated for periosteum and osteocytes and heavily downregulated for senescent cells and tenocytes (Figure 5B). In functional analysis, the top two pathways with major differences in distances between conditions were EPHB and ADGRL (Figure 5E). Conversely, the information flow shows that the majority of pathways are enriched in the vehicle and very few in VD. Notably, resorption pathways are upregulated by VD (RANKL, TNF, MMP, IL6) and downregulated mainly for chemotaxis (CXCL, PTN, CD34) (Figure 5H).

In the bone, vitamin D treatment upregulates interactions involving cortical parenchymal bone cells, such as periosteum, osteocytes, and osteoblasts, while interactions involving fibroblasts, senescent cells, and chondrocytes are predominantly downregulated. Receiver interactions are notably increased for periosteum and osteocytes but markedly reduced for senescent cells and tenocytes (Figure 5B). Functional analysis highlights EPHB and ADGRL as the pathways with the greatest differences between conditions (Figure 5E). The information flow reveals that most pathways are enriched in the vehicle group, with few enriched under vitamin D. Notably, resorption pathways, including RANKL, TNF, MMP, and IL6, are upregulated under vitamin D, while chemotaxis-related pathways, such as CXCL, PTN, and CD34, are downregulated (Figure 5H).

## 4. Discussion

### 4.1. Preprocessing

Cell distribution and quantity across different lineages in different tissues can bring important clues to start single-cell analysis. For instance, the differentiation pattern seen in the cell-type UMAP plot of skin tissue (Figure 1B) is concordant with the current literature that vitamin D promotes keratinocyte differentiation [36,37]. This has relevant implications in the aging process, since impaired differentiation triggers decreased permeability barrier function, which leads to more cutaneous inflammation and ultimately aged related disorders [38,39]. This may lead to a translational approach, like in the use of a calcipotriol cream in the treatment of psoriasis [40], but with an aging approach.

Interestingly, Hu et al. show that VD promotes keratinocyte differentiation, in part at least, by inhibiting the β-catenin pathway, as well as other authors [41,42,43]. Accordingly, Figure 4F demonstrates that skin barrier establishment module is upregulated by vitamin D.

### 4.2. Functional Enrichment

Firstly, to see if vitamin D affected cellular senescence across different tissues, the approach was to evaluate how much of the senescent gene sets were present with or without vitamin D. The results varied according to the senescent gene set used (Appendix A). However, SENmayo has shown to be an optimal candidate for senescent-cell identification using gene signatures [20,44]. Knowing this, the general trend was a downregulation of cellular senescence by vitamin D in prostate and skin and an increase in bone. The current literature suggests that the role of vitamin D is to prevent and rescue senescence phenotypes in fibroblasts, bone mesenchymal stem cells, smooth muscle, endothelium, and chondrocytes [12,45,46,47,48]. Contrastingly, in Appendix A, in bone, chondrocytes were enriched for senescence with VD. Delving deeper, communication between senescent chondrocytes and the other cell types involved mainly extracellular matrix remodeling (collagen deposition). To further corroborate these results, most perturbed modules in the GRN were reviewed for chondrocytes (Figure 3B). Interestingly, most downregulated ones mainly involved matrix remodeling, chondrocyte differentiation, oxidative phosphorylation, and protein endoplasmic reticulum retention. Upregulated ones were about apoptotic immune clearance, glutathione synthesis, and negative regulation of apoptosis. Thus, the apoptotic nature of the chondrocytes promoted by VD may contribute to the senescence profile of these chondrocytes. This aligns with current medical evidence, which indicates that while vitamin D does not significantly impact osteoarthritis progression, it can alleviate pain in vitamin D-deficient patients [49]. This underscores a critical point: clinicians must thoroughly understand the biological rationale of a drug before designing clinical trials based on associations, emphasizing the need for precision medicine approaches to better tackle chronic degenerative conditions.

Cell types that had a reduced senescence score with VD were prostate immune cells, bone adipocytes, and suprabasal 1,2 keratinocytes. Vitamin D’s immunomodulatory effects are well described in the literature [50,51,52], which is in concord with these findings since adipocytes also play an important role in systemic inflammation via adipokines. Accordingly, in skin, as discussed previously, VD upregulates keratinocyte differentiation, which may intervene with the senescence phenotype and ultimately inhibit it.

### 4.3. Gene Regulatory Network Analysis

Secondly, to see how vitamin D affected cellular senescence across different tissues, the angle we took was to perform GRN of the datasets and a differential module eigengene analysis to see how vitamin D affected the networks. We then identified the genes in each module that were involved in senescence with an overlap analysis between the module genes and the senescent gene signatures. Afterwards, we investigated what the relevance was of these overlapped genes in the GRN. Finally, we inquired about the biological processes in which the modules were involved, focusing on the ones that included the overlapped genes.

The first thing we noticed was that almost all modules with the most senescence genes in all the networks were overall downregulated by VD (Figure 3). This has interesting implications since vitamin D has been associated with anti-aging effects [53,54,55]. This positions vitamin D as an interesting candidate for further investigation as a therapeutic agent to promote a healthy aging.

In the prostate network, the main findings were that vitamin D upregulated the majority of the modules in myeloid leukocytes (macrophages and monocytes), and it downregulated them in fibroblasts and diminished their overall expression. The current literature suggests that vitamin D has anti-inflammatory properties in the macrophages [55,56], which agrees with our findings in the way that the blue module (involved with inflammation) is downregulated in almost every cell type. However, questions arose as one of the main processes in the blue module involved negative regulation of myeloid leukocytes, meaning that macrophage activity was upregulated by its inhibition. On further inspection, gene differential expression of the genes of this process revealed that they were actually upregulated by vitamin D, particularly Spi1, involved in macrophage lineage commitment and survival [57,58] and suggesting an overall anti inflammatory effect of vitamin D. Moreover, as chronic inflammation is considered a hallmark of aging [59], here VD may counter this pathological state.

Interestingly, the most downregulated module by VD was the red one, which is involved in matrix reorganization in fibroblasts. This result agrees with the seen inhibitory effect of vitamin D in the cancer-associated fibroblast profile [60,61]. Finally, the turquoise module was the module with the most senescence genes. This module was involved in mitophagy, the growth hormone (GH) receptor, and splicing. VD upregulated this module in macrophages and downregulated it in fibroblasts. This could be due to the fact that VD upregulates an anti-inflammatory profile via bioenergetic alterations, mainly involved with the control of the GH and mitophagy of worn out mitochondria in fibroblasts [62], and promotes an immunomodulatory profile in macrophages. This process may also alleviate another hallmark of aging, which is mitochondrial dysfunction.

Notably, prostate tissue had the senescence genes with the higher connectivity, even considering that this is from PTEN knockout mouse tissue or, in other words, a precancerous model. This highlights the crossroads between aging and cancer, since they share many common mechanisms like genomic instability, telomere attrition, epigenetic alterations, immune dysregulation, mitochondrial dysfunction and cellular senescence [63]. This also points out that vitamin D does influence the gene regulatory networks of both mechanisms significantly.

In the bone network, there are some senescence modules upregulated by VD. For instance, salmon involved the IFN response. Delving deeper, in this biological process, the Bst2 gene turned out to be the most differentially expressed between conditions, and it was downregulated by VD in adipocytes mainly. This may explain how the adipocytes were downregulated by VD in the senescence gene signatures. For the overall trend of upregulation, this may be explained by the hypothesis developed by Newmark [64] in which they explain the link with VD and IFN. Briefly, the vitamin D receptor (VDR) evolved to be a key regulator of immunity via sterol network control, and another key player in this network evolved to be IFN in vertebrates, so the upregulation of one promotes the upregulation of the other, agreeing with our current results.

For the downregulated modules, interestingly, the module with the most senescence genes (turquoise) was involved in negative transcription under stress response, implying that VD upregulates stress transcription via c-JUN activation, which is a key for the anti-proliferative effects of vitamin D [65,66]. Another interesting module, although not included in the overlap analysis, is the green module, which remains as a very significant module in the network.

Interestingly, this module is involved in Signal Recognition Particle (SRP) pathways and is heavily upregulated in all cells, suggesting a strong interaction of VD with this SRP pathway. The SRP pathway is very important in the control of misfolded proteins, so it is very involved in diseases associated with this pathophysiology, like neurodegenerative diseases, autoimmune myositis, and cancer [67]. Remarkably, in the work of Kimmel et al. on the single-cell transcriptomic analysis of aging in murine tissues, they mention that a common pathway involved in aging across tissues was the SRP pathway, underscoring the importance of this pathway in aging [68]. This is in concord with the current research that VD plays a major role in bone aging [10]. However, its definitive function remains elusive since the recent VITAL clinical trial has shown no significant reduction in fractures in non-deficient or osteoporotic patients after VD supplementation [69].

In the skin network, interestingly, work from Segaert suggests that VDR is actively expressed in cycling cells, and when arrested, VDR is decreased [70]. This suggests that the downregulation of gene modules may be actively linked with VDR expression, particularly in senescent cell genes, which may hint at the downregulation of senescent cells by VD to promote this effect on cycling cells, while for upper-level keratinocytes, the literature supports the VD role in the differentiation of these cells [71], concordant with the results (Figure 3C). Another interesting result is that the downregulation of the green module (which has genes involved in inflammation pathways) is preserved across almost every cell type (Figure 3C), agreeing with VD immunomodulatory effects. All the biological phenomena regulated by VD agree with the current literature. For example, the downregulation of oxidative phosphorylation in the yellow module, except in the epidermis (basal and suprabasal 2 cells), determines the stress induced by UV light [72,73] and the inhibition of protein and mRNA synthesis in the brown and dark grey modules, suggesting the energy-conserving profile of vitamin D, which is particularly important in the context of an anti-inflammatory profile.

Interestingly, upregulation of skin barrier function appears in the blue module, and ECM disassembly appears in the purple module’s suprabasal 1 cells; this supports the barrier function of VD in skin and the effect of VD on the hair follicle [74] since the most upregulated cells in the blue module were from germ layer 2. Another remarkable finding is the inhibition of autophagy in the light cyan module, except in the suprabasal cells of the epidermis and hair follicle; this may be because VD protects the cells in charge of maintaining the skin barrier (the suprabasal cells) but does not protect the rest of the tissue to avoid excessive energy consumption in other non-vital sites for barrier function (and thus avoid energy-demanding processes like inflammation). Altogether, vitamin D proves useful to maintain skin barrier function, provide anti-inflammatory effects, and overall to conserve energy for vital skin processes only. These main effects underscore the importance of vitamin D in cellular senescence and aging, since photoaging and inflammaging are crucial to developing pathological aging and cellular senescence since VD modulates the “senescence-associated secretory profile” (SASP) via anti-inflammation, ECM regulation, and energy-conserving actions. So, vitamin D seems to be an attractive approach to aesthetic medicine and cosmetics [38]. Future studies surveying photoaging (like a well-structured cohort) are much needed to assess its proper use and develop clinical trials to validate the findings.

### 4.4. Cell Communication

For wrapping up our analysis, we constructed a cellular communication network to see how senescent cells affected each other and other cell types in each tissue.

In the prostate network, the main findings indicated a reduced senescent fibroblast source—senescent macrophage receiver crosstalk in Figure 4A. Under further inspection, the main pathways involved in this communication involved collagen, agreeing with the information flow of this pathway seen in Figure 4G and the reduced module expression of the fibroblasts in Figure 3A,D. However, a remarkable pathway was the APP one, since in the vehicle, the fibroblasts express APP and the macrophages receive them via Cd74, but the APP pathway in the vitamin D group was nowhere to be found. While there is no evidence about the role of vitamin D on APP expression in the prostate, APP has been shown to promote androgen-dependent growth of cancer [75] and metalloproteinase expression, promoting tumor metastasis [76]. So, this may suggest that VD attenuates prostate cancer through androgen-dependent inhibition and cancer cell migration via APP. Other interesting findings between prostate senescent cells is that IL-6 and IL-10 are enriched in VD senescent cells. This suggests that these cells promote cellular senescence via SASP induction in the prostate and promote an anti-inflammatory shift and matrix reorganization. This may be a protective effect since the activation of IL-6 can be useful to prevent cancer progression via senescence induction [77].

For the communication between senescent cells and the other cell types, we noted a significant downregulation of the sender communication of the top tenth percentile of senescent cells in the VD group. This suggests that VD downregulates the SASP, which agrees with the current literature [12,78]; however, under further inspection and in accordance with our previous findings with IL-6, TGFb and TNF were upregulated by VD in the senescent cells sender profile. This triggered the investigation of these pathways, revealing that TGFb was shifted toward fibroblasts with high CXCL9, while TNF remained targeted mainly to macrophages and senescent cells in both groups. This was interesting, since CXCL9 fibroblast communication was overall upregulated by vitamin D, as seen in Figure 5A. These findings suggest that VD modulates SASP to improve immune function and recruitment via TNF, MIF, and CSF pathways and CXCL9 fibroblast/macrophage interactions, but overall, it reduces SASP, which is reflected in reduced collagen pathways, as seen in Figure 5G.

Another interesting finding was the one with MHCI underrepresented in VD. We investigated the pathway and discovered that proliferating CD8-positive cells were the only receivers in the control group, and they were absent in the VD one. This could be explained by VD’s immunosuppressive and anti proliferative characteristics [79]. So, in conclusion, VD seems to modulate SASP to promote cellular senescence, immune recruitment, and the function of myeloid leukocytes, downregulating fibroblast activity (although not CXCL9 immune functions) and thus diminishing ECM deposition and enhancing senescence profiles in existing senescent cells. This could be useful in early prostate cancer stages; however, in more advanced cases it could prove deleterious. This goes according to the literature where vitamin D deficiency has been associated with higher prostate cancer risk [80] and the same goes for other cancer types, as seen in the vital sub-analyses [81].

For the bone network, in senescent cell communication we found a marked increase in pericyte receiver information and an increase in the rest of the communication of the senescent cell types, particularly in chondrocytes (Figure 4B). On further examination, we found that senescent chondrocytes exhibited a curious communication profile, having some pathways like pre-angiotensinogen, melanocortin, and apelin with some information flow (Figure 4H). While there is some evidence of adropin’s role in alleviating aging in neurodegeneration [82], there is a lack of evidence regarding the role of adropin in cellular senescence. With apelin, there is information on alleviating angiotensin-induced senescence in the endothelium via AMPK and SIRT1 activation [83]. This agrees with our communication network, since apelin from chondrocytes is sent to the endothelium. This suggests that vitamin D attenuates cellular senescence, although the concentrations were high, which could raise the senescence signatures, as seen in Figure 2B. Interestingly, transcriptomic changes can arise with dosages as small as 50 ng injected intraperitoneally [18], although more systemic responses have been observed at 2.5 micrograms per kilogram [84].

Another interesting finding was the downregulation of NOTCH in the vitamin D group, as seen in Figure 4H. This pathway has been shown to induce secondary senescence in a juxtacrine way [85]. Moreover, NOTCH inhibition in mice via myosin light-chain 3 maintenance has been shown to inhibit chondrocyte senescence and osteoarthritis [86]. Thus, vitamin D inhibition of NOTCH seems useful to delay senescence in cartilage.

In the communication between senescent cells and the other cell types, we noticed that parenchymal-cell interactions (mainly periosteum and osteocytes) were upregulated, while fibroblasts, chondrocytes, and senescent cells were downregulated (Figure 5B). To further examine this characterization, we reviewed the interactions between the vitamin D group and we noticed an increase in SASP expression, like IL-6, MMP, and TGF-b (Figure 5H). Other pathways like SPP1, RANKL, and netrin suggest that VD at high concentrations promotes SASP production in osteocytes, osteoblasts, and periosteum mainly to promote the resorption process, while the SASP in other cell types was attenuated since they do not participate in bone resorption, although some pathways remained active, like GAS and APP. This agrees with studies which mention that high doses of VD can promote loss of bone mass density and resorption [87,88], and this may suggest that VD modulates SASP to play a role in resorption. GAS and APP maintenance in non parenchymal cell types from senescent cells could play a role in bone homeostasis promoted by vitamin D, promoting bone formation [89,90]. This may be a new route by which vitamin D prevents the onset and progression of osteoporosis and can be proven key in developing new drugs due to a deeper understanding of bone homeostasis and how cellular senescence could affect it.

Finally, for the skin network, we discovered that communication between senescent cells was hampered in the vitamin D group, as seen in Figure 4C and in the downregulation of the main SASP pathways like MMP, IL1, TGFb, GAS, and CCL in Figure 4I, concordant with the current literature [12]. To further delve into these interactions, we studied, in Figure 4I, the main pathways altered by VD. Integrating this information, the first thing that we noted was the upregulation of fibroblast communication and the shift from collagen, tenascin and gap ECM components to desmosomes and cadherins. This suggests that VD modulates SASP for skin repair via epithelium regeneration rather than fibrotic wound repair, which agrees with other authors’ results [91].

Interestingly, important contributors to the communication network were the germinal layers and the dermal papilla fibroblast cells, which is concordant with the current literature on the importance of VD in hair growth [92]. Remarkably, this study by Joko et al. corroborated the importance of VD via VDR K.O. in the removal of senescence, but in this instance in hair follicle generative layers specifically. However, senescence may have another role in this context of UV radiation, since germinative layer 2 senescent cells with vitamin D were overly expressing ANGPTL4, which has been associated with skin regeneration in wound processes [93]. Another interesting finding in Figure 4I was the upregulation of the galectin pathway in the VD group. Investigating this pathway more deeply, we found out that the main cells receiving information were the macrophages. In the literature, galectin is known to enhance macrophage phagocytic function [94] and polarize macrophages toward an immunomodulatory profile (M2) [95] which could potentiate skin regeneration via vitamin D SASP modulation.

Subsequently, we investigated the senescent cell communication with other cell types and the overall changes of vitamin D in radiated skin. We noticed that the senescent cells were the most altered source and T cells the most changed receiver in the network (Figure 5C). Further investigating both phenomena, the main pathways sent from senescent cells involved matrix remodeling as the one seen in Figure 4I, concordant with what is seen in the overall pathway differential analysis in Figure 5I. Interestingly, all the fibroblasts were included in the senescent-cell group, reflecting the fact that all fibroblasts went in the 10% of skin cells with the highest enrichment SENmayo score. This may highlight the importance of cellular senescence in fibroblast-mediated skin repair [96] and the way in which vitamin D modulates its secretome to profile a dermal regeneration phenotype rather than a fibrotic process, which agrees with other works on the subject [91].

Another interesting approach was the notable upregulation of ANGPTL seen in Figure 5I. We observed that the only sources for this pathway in the VD group were the sebaceous glands, the upper hair follicle basal cells, and the germinative layer cells. Concurrently, according to the work from Dahlhoff et al., ANGPTL4 proves to be important in reducing lipid droplet size and mass, suggesting ANGPTL4 mimics therapeutic agents for acne [97], and thus, this could provide some evidence for VD as a possible relieving agent in acne patients via ANGPTL activation [98]. For the role of ANGPTL and the hair follicle, various studies have shown that this pathway promotes follicle regeneration and hair growth via angiogenesis induction; thus, this could prove useful to promote VD as a possible treatment for hair follicle diseases like alopecia and acne.

While scRNA-seq is a powerful tool for studying aging, it has key limitations that necessitate complementary approaches. One major constraint is its ability to provide only single snapshots of cellular states, which makes it difficult to capture the dynamic progression of aging over time, such as the gradual decline in immune function [99]. Furthermore, scRNA-seq often struggles to capture isoforms and the complete transcriptome landscape due to low sequencing depth, limiting its ability to fully represent complex gene expression patterns [100]. Additionally, scRNA-seq often isolates individual tissues, missing critical inter-organ interactions, such as those linking metabolic dysfunction in the liver to systemic inflammation. Its focus on transcription alone also excludes other key aging factors, including protein damage and epigenetic changes, which are essential to understanding cellular senescence [100,101].

Reliance on murine models complicates the translation of the scRNA-seq findings to human aging due to species-specific differences in aging processes and molecular pathways, particularly in immune and cardiovascular systems [102]. Accelerated aging models in mice often produce artificial molecular signatures, such as exaggerated DNA damage and oxidative stress responses, that diverge from natural human aging [102,103]. Even naturally aged mice show limited overlap with human transcriptomic aging markers, particularly in brain and heart tissues, due to evolutionary divergence [99,102]. Additionally, their short lifespan prevents studies of late-life phenomena like neurodegeneration and frailty, crucial to human aging research [102]. These challenges highlight the need for validating findings in human tissues and using complementary approaches such as multi-omics and cross-tissue analyses [101].

## 5. Conclusions

Some perspectives and future directions can also be drawn form this work. Among these, we can mention the following:Vitamin D seems to play a role in senescent cell expression, particularly in the SASP profile, given the most significant change in SENmayo scores and cellular communication relevance in senescent cells across the prostate, bone, and skin.Gene regulatory networks of the three tissues show that almost all the modules with the most senescent genes were downregulated by VD, suggesting the importance of VD in senescent profiles and in aging.The role of VD in the SRP pathway in bone reveals trailblazing insight into aging bone, pointing out the relevance of future investigations to further characterize these findings.The maintenance of cellular senescence by vitamin D via IL-6, APP reduction, matrix reorganization, and macrophage immunomodulation seems useful for early stages of prostate cancer but inconclusive for later stages.Senescent chondrocyte profiles triggered by VD could provide clues to the role of these cells in energy homeostasis in bone.Paracrine senescence seems to be inhibited in bone by vitamin D via NOTCH downregulation.ANGPTL and galectin promotion by vitamin D in senescent cells may play a crucial role in the radiated skin repair process via lipid metabolism and immune clearance.The senescent fibroblast SASP profile in irradiated skin appears to be relevant in extracellular matrix remodeling, and vitamin D seems to modulate it towards an anti-fibrotic type.Further validation of these findings with other experimental methods is crucial for further understanding the role of vitamin D in cellular senescence.

## Figures and Tables

**Figure 1 nutrients-17-00429-f001:**
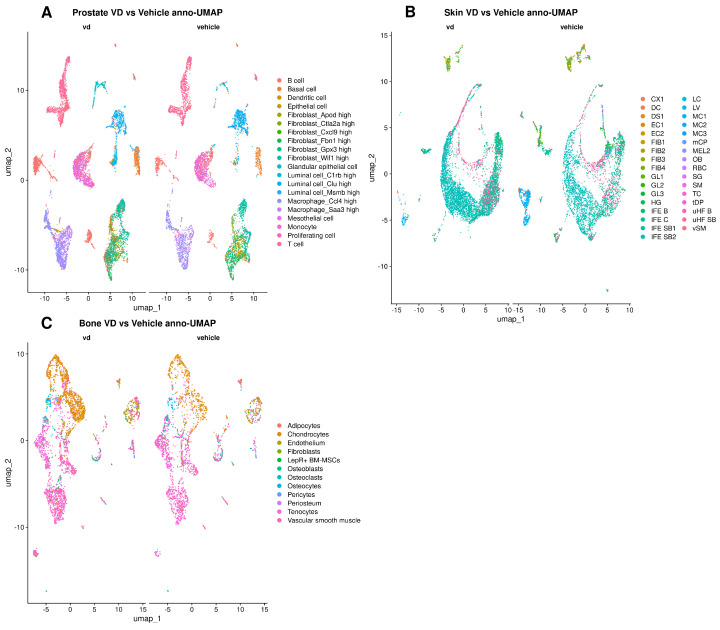
UMAP plots of vitamin D vs. vehicle in mouse prostate (**A**), skin (**B**), and bone (**C**). Each point represents a single cell, color coded by cell type (see legend). UMAP (Uniform Manifold Approximation and Projection) arranges cells with similar gene expression closer together. For each tissue, cells from vitamin D treated mice (“VD”) appear on the left and those from vehicle treated mice (“vehicle”) on the right. Changes in clustering patterns between the two groups highlight how vitamin D may alter the distribution of these cell types.

**Figure 2 nutrients-17-00429-f002:**
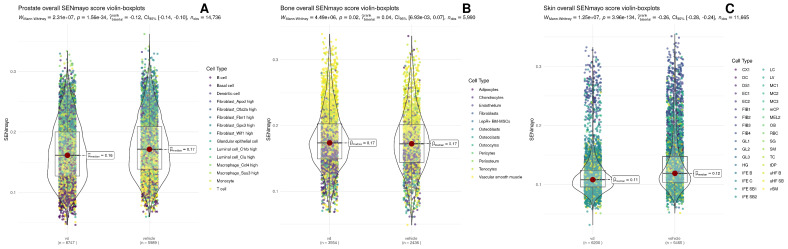
SENmayo enrichment scores in VD vs. vehicle across prostate (**A**), bone (**B**), and skin (**C**). The y axis measures how closely each cell’s gene expression matches a senescence related signature (higher scores mean stronger “senescence like” profile). Each violin/box plot shows the distribution of cells (dots) color coded by cell type, with the box plot marking the median and interquartile range. In the prostate, macrophages and fibroblasts typically score higher than B and T cells. In bone, vascular smooth muscle and tenocytes trend higher than adipocytes. In skin, suprabasal cells make up the bulk of the distribution, while fibroblasts and dendritic cells occupy the upper range.

**Figure 3 nutrients-17-00429-f003:**
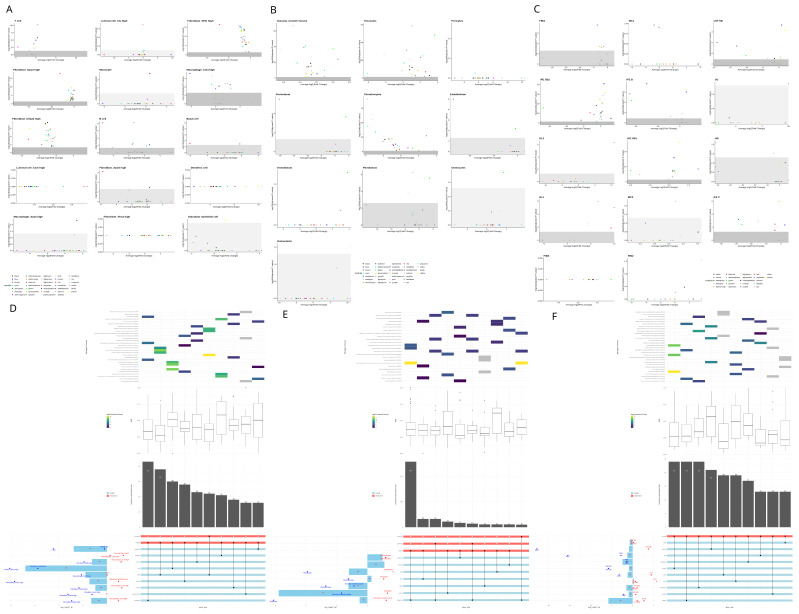
Gene Regulatory networks in prostate, bone, and skin. Panels (**A**–**C**) show volcano plots of differential module eigengene expression (vitamin D vs. vehicle) for each cell type in prostate (**A**), bone (**B**), and skin (**C**). The x-axis indicates log fold change, and the y-axis represents statistical significance; modules appearing farther to the left or right and higher on the y-axis are more strongly and significantly altered. Panels (**D**–**F**) present complex UpSet plots illustrating how gene modules (blue rows) intersect with senescence-associated genes (red rows). From bottom to top in each UpSet panel, the intersection chart shows which sets overlap; the barplot above it indicates the intersection size (larger bars mean more genes); the box plot displays the average connectivity (how strongly those genes are connected to other genes); and at the top, a heatmap uses a color gradient from purple (low) to yellow (high) to show the combined GO enrichment score, a value that reflects both how much each biological process is associated with the overlapped genes. On the left side of each UpSet panel, a barplot shows the module log fold change (vitamin D vs. vehicle), with points marking the cell types that have the highest and lowest log fold change. The longer the bar, the greater the change in module expression under vitamin D, with bars above zero indicating higher expression (upregulation) and bars below zero indicating lower expression (downregulation). Red asterisks on the bars indicate if the log fold change was significant (*p* < 0.05). These analyses are collectively called differential module eigengene (DME) expression. Together, these results show which gene modules are most affected by vitamin D in each tissue and in which cell types, particularly those overlapping with senescence-related genes, highlighting potential pathways of interest for further study.

**Figure 4 nutrients-17-00429-f004:**
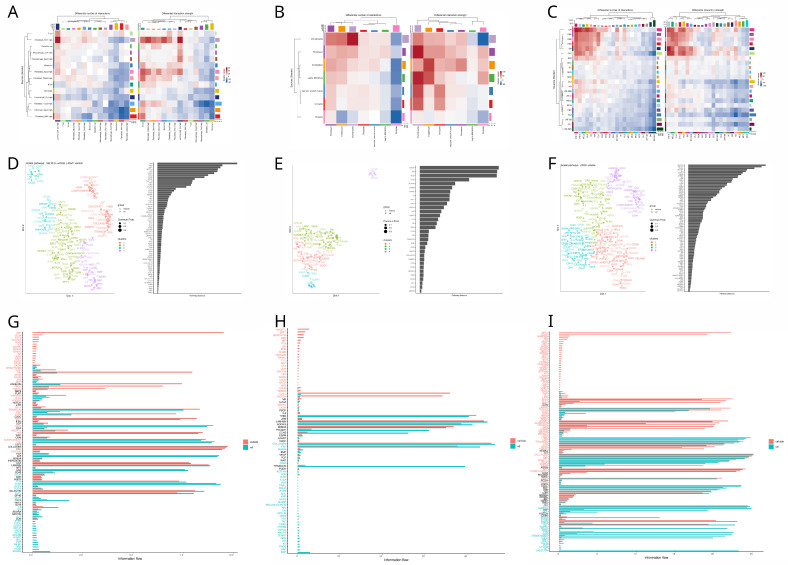
Cellular communication among senescent cells in prostate, bone, and skin under vitamin D vs. vehicle treatment. Heatmaps in panels (**A**–**C**) illustrate how senescent cells in each tissue (prostate in (**A**), bone in (**B**), and skin in (**C**)) alter their intercellular communication under vitamin D relative to vehicle, with the left heatmaps showing the difference in the number of interactions and the right heatmaps showing the difference in the overall interaction strength. Hierarchical clustering along the axes highlights which senescent cell populations exhibit the most pronounced changes; warmer colors (red) indicate greater interaction shifts under vitamin D, while cooler colors (blue) reflect reduced crosstalk. Panels (**D**–**F**) depict scatter plots that map signaling pathways (colored by functional class) based on their similarity, with greater distances suggesting more divergent functional profiles between vitamin D and vehicle. The adjoining bar charts rank these pathways by their average distance (or divergence), revealing which pathways in senescent cells undergo the most significant shifts under vitamin D. Finally, panels (**G**–**I**) present barplots summarizing the top differentially regulated signaling pathways in senescent cells for each cell type, with the x-axis showing “information flow” (a measure of how strongly a pathway contributes to cell-cell communication) and bars color-coded to indicate stronger activity under vitamin D (red) versus vehicle (blue).

**Figure 5 nutrients-17-00429-f005:**
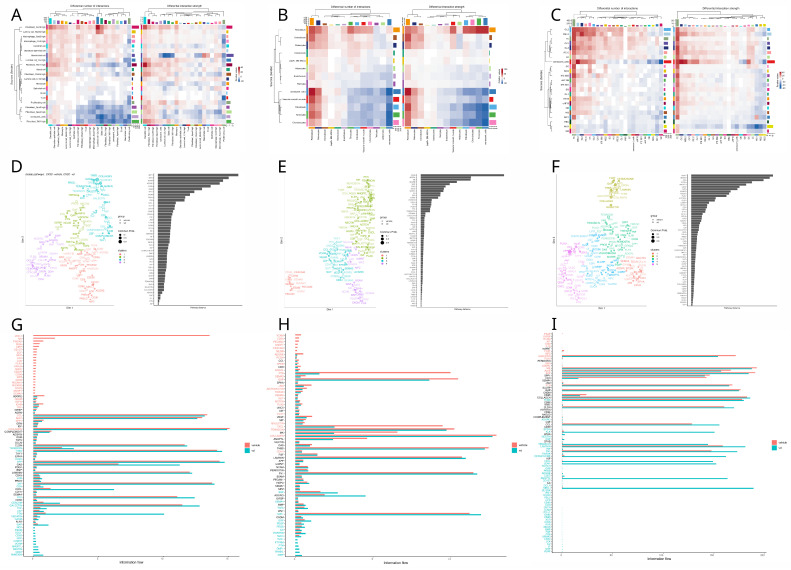
Cellular communication between senescent cells and other cell types in prostate, bone, and skin under vitamin D vs. vehicle treatment. Heatmaps in panels (**A**–**C**) illustrate how each tissue (prostate in (**A**), bone in (**B**), and skin in (**C**)) alters its crosstalk between senescent cells and other cell types under vitamin D relative to vehicle, with the left heatmaps showing the difference in the number of interactions and the right heatmaps showing the difference in the overall interaction strength. Hierarchical clustering along the axes highlights which cell populations exhibit the most pronounced changes when engaging with senescent cells; warmer colors (red) indicate greater interaction shifts under vitamin D, while cooler colors (blue) reflect reduced crosstalk. Panels (**D**–**F**) depict scatter plots that map the functional similarity of key signaling pathways based on how differently they operate in the vitamin D versus vehicle groups, with greater distances corresponding to more pronounced treatment effects. The adjacent bar charts rank these pathways by their average distance (or divergence), pinpointing those most influenced by vitamin D in senescent-cell interactions. Finally, panels (**G**–**I**) display barplots summarizing the top differentially regulated signaling pathways in each cell type, with the x-axis denoting “information flow” (a measure of how strongly each pathway contributes to communication with senescent cells) and bars color-coded to indicate higher activity under vitamin D (red) or vehicle (blue).

## Data Availability

All three datasets used for analysis were retrieved from the Gene Expression Omnibus database (GEO), being prostate code GSE164858, bone GSE220836 and skin GSE173385. For additional information regarding the datasets, it is highly recommended to check the prostate (Abu el Maaty et al., 2021 [16]), bone (Hanai et al., 2023 [17]) and skin (Lin et al., 2022 [18]) source articles. For the senescence associated gene scores, they are publicly available in the web. Then, SENmayo can be downloaded from the first supplemental file of their article (Saul et al., 2022 [20]), Geneage from the url https://genomics.senescence.info/genes/models.html (accessed on 15 January 2025) and Cellage from the url https://genomics.senescence.info/cells/ (accessed on 15 January 2025). Finally, regarding the annotation references, prostate reference DGE and cell annotations can be downloaded from the cell mouse atlas web page (https://bis.zju.edu.cn/MCA/gallery.html?tissue=Adult-Prostate (accessed on 15 January 2025)), skin reference from GEO with code GSE129218, and bone reference from the skeleton cell atlas (https://www.skeletalcellatlas.org/ (accessed on 15 January 2025)) as a loom file. For the direct download links, see the metadata directory with the csv containing the URL’s for each input. All code is available in the github repository https://github.com/Ssdzem/VD-sc-senescence (accessed on 15 January 2025).

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
