# Peer review of "Single-Cell Analysis Dissects the Effects of Vitamin D on Genetic Senescence Signatures Across Murine Tissues"

_nutrients, 2025, doi:10.3390/nu17030429_

Round 1
Reviewer 1 Report
Comments and Suggestions for Authors
Reviewers’ comments on MS Emilio Sasa-Dias et al., Nutrients
In general: the question is relevant and interesting.
However, the conceptualization is not clear enough. The authors should justify why did they choose the three mentioned tissue (skin, prostata and bone) – and the whole tissue or only special cells from these tissues?
In general: the exact question, the accurate aim of the study is missing or should be more precise (at the end of the introduction).
They use some strange terms, just like in the introduction „non-communicable diseases” – they should write more concise and clear. in general, corrections of a native speaker would improve the quality of the MS.
There are some missing citations – there are several statements without justifying literature, citations.
Questions about the methods:
You do not measure upon a certain protocol, however, You used and reevaluated other measurements from the literature – am I right?
There is no data on search keys, inclusion and exclusion criteria and why did they exlude 2 studies and included 4 studies for evaluation.
UV-irradiated skin data did not influenced the results?
Give data on vitamin D levels in deficient and „normal”/ supplemented groups. Were it similar in the different studies? If it is not known, it is a bias in evaluating differences of gene expression data.
From chapter 2.4.2. references are given without numbers. Before itt he authors give reference numbers from 1-24 and these are given at the end, in the reference list. Following 2.4.2. they give the first author and the year of publication and these references are not mentioned at the end, in the refereces. Perhaps this is not the final version of the MS which was uploaded, because a lot of points of the MS seems to be not finalized. They should check it more carefully.
Statistical methods seems to be adequate.
Description of the methods seems adequate, however, there are some problems with the language, run a language check again, please.
One important completion needed: give the minimal and maximal logfoldchange in 2.6.2.!
Results:
Difine all the axes! They are not clear (Fig 1., 2.)
The authors sometimes write about cells/ transcriptom analysis (Fig 1-3), later exosomes (Fig. 4-5) – however, in the methods do not mention evaluation of exosomes.
Results are a bit confusing. Write every results clearly – we should undestand every figure in its’ own upon the figure and the figure legend. Correct the figures and the legends, also!
For the discussion would be interesting, from which vitamin D level can be seen the transcriptomic changes? You mention Vital trial, which seems to be the perfect example, because following 2000 IU vitamin D supplementation the patients might have similar vitamin D levels like controls which diminished the difference between control and treated patients – there is not known vitamin D levels in this study to answer this question. You should exclude the same bias – so You should give the vitamin D levels for your groups.
Theoretical part of the MS and the discussion is strong, however, clinical implications should be improved. Why is it important for the clinicians?
In general: the idea of the study is very promising and good. In its present form there is so much mistakes, I hope, I mention the most important ones – a very careful final check up would be necessary.
Comments on the Quality of English Language
Extensive reviosion of the language would be useful.
Author Response
Dear Reviewer 1,
To clearly address each of your kind observations, we have made a point-by-point list containing your comments and suggestions as well as our responses (in bold type to ease reading). Changes are referenced by the text lines in the revised manuscript. Thank you for your quite professional reading of our manuscript and for proposing changes that no doubt have greatly improved its scope and readability.
General questions
The authors should justify why did they choose the three mentioned tissues (skin, prostate and bone) – and the whole tissue or only special cells from these tissues?
R= The main reason was that these tissues were the only ones that met our inclusion criteria (lines 79 to 81).
Nonetheless, we put the reader in context by exposing why there are very few experiments using this technique (line 55 to 57) and why these tissues are important in VD metabolism and aging (lines 50-60 and 66-68).
The accurate aim of the study is missing or should be more precise (at the end of the introduction)
R= The end paragraph of the introduction (lines 70 to 76) was rewritten to be more clear and precise:
“In this study, we investigate the relationship between vitamin D and cellular senescence using single-cell RNA sequencing, aiming to understand how vitamin D influences senescence across various tissues and cell types. By taking a non-canonical perspective on the aging process, we seek to reveal novel insights into how vitamin D might delay or modulate age-related changes. Ultimately, this work may inspire new research and potential interventions against aging-related diseases by harnessing vitamin D’s beneficial effects.”
They use some strange terms, just like in the introduction "non-communicable diseases” – they should write more concise and clear. In general, corrections of a native speaker would improve the quality of the MS.
R= A full language screening was carried through to ensure the clarity of the work, it was also copy-edited by a professional academic proofreader. Thanks for the callout.
There are some missing citations – there are several statements without justifying literature and citations.
R= There was a mistyping when changing to LaTeX typesetting and some citations were lost. We have solved this and now all the citations are present.
Questions about methods
You do not measure upon a certain protocol, however, You used and reevaluated other measurements from the literature – am I right?
Thank you for your comment. To clarify, in our study, we used publicly available data from three different experiments. By analyzing these existing datasets, we were able to gain valuable insights without conducting new experiments. Using this approach ensures our results are based on well-established data. It also helps us build on previous research and share new perspectives with the scientific community. We've updated the manuscript to make this point clearer.
There is no data on search keys, inclusion and exclusion criteria and why did they exclude 2 studies and included 4 studies for evaluation.
R= By search keys you mean MESH terms? Since the keywords were provided in the first sentence of the data acquisition section. As you likely recall Medical Subheading (MESH) terms form a standardized set of keywords (formally, an ontology) that allow for systematic and prioritized searches. This is now clarified in the manuscript.
For the inclusion and exclusion criteria, we added the inclusion, exclusion and elimination criteria in lines 79 to 87, rewriting the paragraph as follows:
“A research in literature was made with the keywords “vitamin D” and “single cell RNA seq” in PUBMED and GoogleScholar with the aim to recruit publicly available, single cell RNA seq data of experiments using vitamin D. Six datasets were originally recovered. However, under further scrutiny, two were using bulk rna sequencing techniques, so they were excluded. Four datasets were selected for further data preprocessing (Abu el Maaty et al. 2021; Hanai et al. 2023; Lin et al. 2022; McCray et al. 2021). Nevertheless, data analysis was performed only with the databases of mouse bone, skin and prostate (Abu el Maaty et al. 2021; Hanai et al. 2023; Lin et al. 2022) given quality control results (see Preprocessing section).”
Mainly, with the search we did using the keywords provided, we found 6 experiments that matched the query, However, under further inspection, we found that 2 experiments used mainly bulk seq so they were excluded.
UV-irradiated skin data did not influenced the results?
R= Yes, it was very clear in the enrichment analysis the stress response in both vehicle and VD groups. However this provided insights of the vitamin D action under this situation and particularly how it affects cellular senescence in UV radiated skin. So while a limitation to understand normal physiology, it was valuable for exploring the changes in UV radiated skin.
Give data on vitamin D levels in deficient and "normal”/ supplemented groups. Were it similar in the different studies? If it is not known, it is a bias in evaluating differences of gene expression data.
R= To address this observation, we added a paragraph containing the vitamin D concentrations and experiments structure of each dataset in lines 99 to 110.
From chapter 2.4.2. references are given without numbers. Before itt he authors give reference numbers from 1-24 and these are given at the end, in the reference list. Following 2.4.2. they give the first author and the year of publication and these references are not mentioned at the end, in the refereces. Perhaps this is not the final version of the MS which was uploaded, because a lot of points of the MS seems to be not finalized. They should check it more carefully.
R= Thank you for the observation. AS we have sais it was a computer error in the typesetting process. This has been solved in the revised version of the manuscript.
There are some problems with the language, run a language check again, please.
R= Thanks for the observation. This has been addressed by performing a careful professional language screening and proofreading, specially in the results section.
Give the minimal and maximal logfoldchange in 2.6.2.!
R= It was indeed given, but perhaps not clearly enough. These values are displayed in figure 3 as dots in the barplot to the left of the upset plot. The figure legend of said figure has been updated to address this confusion.
Questions about results
Define all the axes! They are not clear (Fig 1., 2.)
R= In figure 1 the axes in a UMAP plot are arbitrary and do not correspond to specific features of the original data. Instead, they represent abstract embedding dimensions created during the reduction process. The positions of points and clusters on the plot reflect the relative similarities and structures within the dataset, but the axes themselves are meaningless and should not be interpreted. This has been explained in the respective figure legend and in lines 184 to 186.
In figure 2. the y axis represent the enrichment of the gene set, in this case, the senmayo gene set. The enrichment score, which is a measure of how strongly the genes in the SENmayo set are active in the cells. The figure 2 legend has been changed to address this issue.
The authors sometimes write about cells/ transcriptome analysis (Fig 1-3), later exosomes (Fig. 4-5) – however, in the methods do not mention evaluation of exosomes.
R= You are absolutely right, this misleading wording of exosomes has been removed from the article. Thanks.
Results are a bit confusing. Write every results clearly – we should understand every figure in its’ own upon the figure and the figure legend. Correct the figures and the legends, also!
R= We have rewritten all the results section to improve clarity and precision. We also rephrase all figure legends in order to make clear the relevance and interpretation for every figure on its own, only with the figure legend and the figure itself.
Questions about discussion
For the discussion would be interesting, from which vitamin D level can be seen the transcriptomic changes? You should exclude the same bias – so You should give the vitamin D levels for your groups.
R= This topic has been added and discussed in lines 568 to 570. Although there is not much information yet in the literature for properly answering this inquiry. As it is still a debated issue.
Clinical implications should be improved. Why is it important for clinicians?
R= This was addressed in every section of the discussion. Specifically, in lines 363 to 364, 388-393, 412-413, 508-510, 553-555, 591-593 and 637 to 640.
Reviewer 2 Report
Comments and Suggestions for Authors
This study utilizes single-cell RNA sequencing (scRNA-seq) to investigate the influence of vitamin D (VD) on cellular senescence across multiple tissues.
- Please justify why bone, prostate, and skin tissues were selected for this study.
- Please provide the full names for “SASP” and “UMAP.”
- The authors state that “VD supplementation significantly decreased senescence scores in the skin (p = 3.96e-134) and prostate (p = 1.56e-34).” What were the results for bone tissue?
- Please discuss the limitations of murine models and scRNA-seq in accurately reflecting the complexity of human aging.
Author Response
Dear Reviewer 1,
To clearly address each of your kind observations, we have made a point-by-point list containing your comments and suggestions as well as our responses (in bold type to ease reading). Changes are referenced by the text lines in the revised manuscript. Thank you for your quite professional reading of our manuscript and for proposing changes that no doubt have greatly improved its scope and readability.
Please justify why bone, prostate, and skin tissues were selected for this study.
R= The main reason was that these tissues were the only ones that met our inclusion criteria (lines 79 to 81).
Nonetheless, we put the reader in context by exposing why there are very few experiments using this technique (line 55 to 57) and why these tissues are important in VD metabolism and aging (lines 50-60 and 66-68).
Please provide the full names for “SASP” and “UMAP.”
R= The full names are provided at the first time of usage of the acronym in the paper (SASP line 506 and UMAP line 125)
The authors state that “VD supplementation significantly decreased senescence scores in the skin (p = 3.96e-134) and prostate (p = 1.56e-34).” What were the results for bone tissue?
R= This is addressed in figure 2 at the top of the violin plots and also in the text lines 208-210
Please discuss the limitations of murine models and scRNA-seq in accurately reflecting the complexity of human aging.
R= Thank you. This was a much needed section of the discussion and it was added at the end of it (lines 641 to 662)